# Investigating the Mechanical Properties and Durability of Metakaolin-Incorporated Mortar by Different Curing Methods

**DOI:** 10.3390/ma15062035

**Published:** 2022-03-10

**Authors:** Yudong Dong, Lianjun Pei, Jindong Fu, Yalong Yang, Tong Liu, Huihui Liang, Hongjian Yang

**Affiliations:** 1School of Chemical Engineering and Technology, Hebei University of Technology, Tianjin 300130, China; dongyd2022@163.com (Y.D.); 201921503015@stu.hebut.edu.cn (T.L.); 202031504010@stu.hebut.edu.cn (H.L.); 2Tianjin Energy Investment Group Co., Ltd., Tianjin 300050, China; h20220217@126.com (L.P.); fanwq123@126.com (J.F.); 3Tianjin Cheng An Thermal Power Co., Ltd., Tianjin 300161, China; yangyl0121@163.com

**Keywords:** cement, early strength, wet curing time, metakaolin, durability, sulfate erosion

## Abstract

In this paper, the traditional, silicate-based Portland cement (PC) was employed as the control to explore the impact of adding varying amounts of metakaolin (MK) on the mechanical properties of cement mortar. In fact, as a mineral admixture, metakaolin (MK) has the ability to significantly improve the early strength and sulfate resistance of cement mortar in traditional, silicate-based Portland cement (PC). In addition to this, the performance of Portland cement mortar is greatly affected by the curing mode. The previous research mainly stays in the intermittent curing and alkaline excitation mode, and there are few studies on the influence of relatively humidity on it. Moreover, the paper investigated the impact of four different curing methods about humidity on the mechanical properties and sulfate resistance. The results show that the best content of metakaolin in Portland cement is 10% (M10), and the best curing method is 95% humidity in the first three days followed by 60% humidity in the later period (3#). Based on previous literature that suggests that adding MK thickens water film layer on the surface of mortar, the mechanism of MK increasing the early strength of cement was analyzed. The compressive strength of the Portland cement containing 10% MK (M10) after 1 day curing is 3.18 times that of pristine PC mortar, and is comparable if PC is cured for three days under the same curing conditions. The traditional PC mortar is highly dependent on the wet curing time, and normally requires a curing time of at least seven days. However, the incorporation of MK can greatly reduce the sensitivity of Portland cement to water; MK cement mortar with only three days wet curing (3#M10) can reach 49.12 MPa after 28 days, which can greatly shorten the otherwise lengthy wet curing time. Lastly, the cement specimens with MK also demonstrated excellent resistance against sulfate corrosion. The work will provide a strong theoretical basis for the early demolding of cement products in construction projects. At the same time, this study can also provide a theoretical reference for the construction of climate drought and saline land areas, which has great reference value.

## 1. Introduction

Ordinary Portland cement with its stable performance, less pollution, and high yield characteristics has been the most important and the most widely cement used in the building materials market. However, ordinary Portland cement also has the shortcomings of low early strength and late strength, but most of the previous studies only focus on the improvement of Portland cement late strength, rather than the early strength change. However, the early strength of ordinary Portland cement is a very important mechanical property index of cement products, which determines the demolding time, production efficiency and production cost of cement products. The early hydration process (in the first 7 days) of traditional Portland cement is highly dependent on water, and the required wet curing time is very lengthy [1,2]. In the drier areas of the north of China, especially in the northwestern of China region, the dry climate and large variation in the saline land elevates the currently high standard for early wet curing time of buildings and the sulfate resistance [3]. Therefore, it is necessary to find a suitable modifier to maintain the mechanical properties of Portland cement in a low humidity environment. At the same time, because the hydration process of Portland cement is highly dependent on water, it is also very important to explore the influence of different curing methods on it.

Metakaolin (MK) is a category of mineral admixtures that possess a high pozzolanic activity. It is primarily comprised of SiO_2_ and Al_2_O_3_, which can react with the cement hydration product, Ca(OH)_2_, to form the corresponding hydrated calcium silicate gel [4,5,6,7], and thereby enhance the compressive strength of the resulting compound. In addition, MK possesses a strong micro-aggregation effect and nucleation effect, provides more nucleation sites and accelerates reaction rates [8]. The small particle size of MK can easily fill the void gaps within cement mortar, and thereby, increases its density, which further improves the mechanical properties of the mortar compound [9,10,11]. Furthermore, the small particle size of MK enables a large specific surface area and water absorption capacity, which will increase the homogeneity, yet, reduce the free water content within the mortar system [12]. However, the water film on the mortar surface will thicken, which in turn, increases the lubrication between some interfaces, increasing the reaction rate and extent, and further increasing the early strength of cement mortar [6,12,13]. Many scientists globally have confirmed that MK can improve the durability of cement mortar to large degrees [14,15,16]. This improvement can be attributed to the significant depletion of Ca(OH)_2_ by SiO_2_ and Al_2_O_3_ in MK, which results in a weaker alkaline environment and improves the sulfate resistance [17,18]. In addition, saline land contains a large number of sulfate groups, thus, MK incorporation is crucial and significant especially in mortar that are mixed with saline land. In this paper, a Na_2_SO_4_ solution with a mass fraction of 5% was introduced as an erosion solvent to survey the sulfate resistance of the mortar specimen. The change in the mass and compressive strength were employed to gauge the resistance of the specimens to sulfate corrosion.

There are indeed some previous studies on metakaolin improving the performance of Portland cement, but there are not many studies on metakaolin cement mortar under different curing conditions. However, there have been many previous studies on the effect of different curing methods on the properties of cement mortars. For example, Amr I.I.H. used the intermittent curing method to product the class F fly ash-based geopolymer. The result showed that the intermittent curing scheme at 70 °C for 4 steps for continuous 6 h of heat curing in each step followed by 18 h of ambient temperature proved to improve the geopolymer mortar compressive strength at the end of each curing step with no adverse effect on the strength [19]. Hilal E.H. used the curing method of alkali-activated materials to produce concrete. Results showed that the optimum curing regime for alkali-activated blended concrete mixtures made with 0 and 25% fly ash was a combination of water and subsequent air curing, while that for mixes made with 50% fly ash was continuous water curing [20]. Athira et al. studied the influence of ambient, heat, water, and other curing methods on the performance of slag, fly ash and a few other precursors based on alkali-activated binders and showed that a trend of reduction in the compressive strength is witnessed with the increase in the replacement levels of alternative precursor materials with slag/fly ash under heat curing, water curing, and ambient curing [21]. The previous research mainly stays in the intermittent curing and alkaline excitation mode, and there are few studies on the influence of relative humidity on it. Therefore, the paper investigated the impact of four different curing methods about humidity on the mechanical properties and sulfate resistance.

However, most previous studies were carried out under standard curing conditions and did not take into account the arid, low-humidity environment of actual production. Therefore, it is of practical significance to explore the effect of dry environment on cement mortar samples. At the same time, under the condition of dry and low humidity, it is more meaningful to study the influence of different curing methods on the properties of metakaolin cement mortar in practical production.

## 2. Materials and Methods

### 2.1. Raw Materials

The ordinary Portland cement (PO 42.5) used in this research was purchased from Xinyue Brand of Tangshan Tianlu Cement Co., Ltd. (Tangshan, China) Metakaolin was originated from the place of Inner Mongolia (China). The chemical composition of the raw materials is summarized in Table 1. The physical properties of the raw materials are shown in Figure 1. The chemical composition and particle size distribution were measured by X-ray fluorescence (XRF, ARLQUANT X, ThermoFisher Scientific, Shanghai, China) and laser particle size analyzer (LSPA, Anton paar LitesizerTM500, Malvern, Worcs, UK). From the Figure 1, the median particle size (D_50_) of PC and MK are 178.25 μm and 1.589 μm, respectively. At the same time, the BET surface area of PC and MK are 2.6659 m^2^/g and 229.1912 m^2^/g, respectively. After the ordinary river sand was dried, it was passed through a 40-mesh sieve to obtain treated river sand. The physical properties of river sand are summarized in Table 2. The water and superplasticizer needed for the experiment was laboratory tap water and polycarboxylate, respectively.

### 2.2. Specimen Preparation

The cement chosen for casting the test specimens was designed for a 28-day compressive strength of about 43.7 MPa. The mix proportions were 0.45:1:1.5 (water: cement: sand by weight). The cement mortar mixer was used to uniformly stir the precursor materials, which were then sealed in 40 mm × 40 mm × 40 mm molds for natural curing for 24 h. Metakaolin was integrated at 0, 5, 10, 15, and 20 wt%, which were denoted as PC, M5, M10, M15, and M20, respectively. Superplasticizers were added to the mixtures of each ratio. Due to the small particle size of MK, the fluidity of the mortar decreased with the increase of the replacement amount of MK. In order to ensure the workability of the mortar, it is necessary to gradually increase the amount of superplasticizer. The different amount of superplasticizer depends on that amount required to achieve the standard consistency for cement mortar sample [22]. Moreover, three blocks were prepared as replicates at each composition. Each block was prepared as summarized in Table 3.

This paper focuses on addressing the aspect: To explore the influence of metakaolin on the mechanical properties (especially the intensity of the first 3 day and 28 day) and sulfate resistance of Portland cement mortar in each group under different curing methods to provide a feasibility reference for the maintenance of cement products and the timing of mold removal in engineering. Most previous studies were carried out under standard curing conditions and did not take into account the arid, low-humidity environment of actual production. This paper focuses on the effect of different curing methods on the properties of metakaolin-incorporated cement mortar under a dry, natural environment. In the dry areas of northwest China, the relative humidity of the air in the natural environment is generally around 60%. Therefore, a combined curing method is adopted in this paper, that is, cured in the high humidity incubator in the early stage, and then cured in the low humidity (60%) drying incubator. Therefore, this paper proposes four different curing methods (all at 20 °C): 0# (humidity 60%); 1# (Standard Curing, humidity 95%); 2# (the first 7 days: humidity 95% followed by humidity 60% for remaining days); 3# (the first 3 days: humidity 95% followed by humidity 60% for remaining days). This paper sought to explore the performance of the above five groups of samples under each curing method. Three samples were selected from each group of samples for various performance tests, and the average value of the three values was taken as the final test result.

The specimens of each group were cured to the corresponding time, while the compressive strength of each group was measured. After 28 d curing, the durability of mortar specimens in each group was investigated by immersing a (5 wt%) Na_2_SO_4_ solution for the corresponding erosion durations (28 day, 90 day, 120 day, 180 day).

### 2.3. Test Methods

After each group of specimens were cured for the corresponding duration, they were characterized according to the national standard GB/17671-1999 for the compressive strength test [23]. The compressive strength of each specimen was assessed on a pressure testing machine (CMT6104, Sans, Shenzhen, China) with a maximum load of 300 kN and a loading speed of 2.4 kN/s.

The specimens were cured in a constant temperature and humidity incubator for 28 days according to the corresponding curing method above. Three samples were directly used to test the compressive strength after 28 days of curing, and the other three samples were used to test the resistance to sulfate attack. After removal from the incubator, the mass before erosion was measured (m_1_), and then the samples were placed into a (5 wt%) Na_2_SO_4_ solution. During this period, the erosion solution was replaced every 15 days until the corresponding term was reached. Subsequently, the specimen was displaced onto ambient conditions for half a day, and the resulting mass was recorded (m_2_). The mass loss rate on a mortar block was calculated according to Equation (1). After the mass loss rate of the sample is tested, the compressive strength of the samples at the corresponding erosion age can be tested directly, so that the mass loss rate and compressive strength at different erosion ages can be obtained.
(1)$$\Delta m=\frac{m_{2}{\;-\;m}_{1}}{m_{1}}\;\times \;100\%$$

To determine the crystal phase composition of the hydration products, X-ray diffraction analysis (XRD, Bruker D8 Discover, Karlsruhe, Germany) with Cu Kα radiation was employed to appraise each specimen with Cu Kα radiation. This included: setting scanning step 0.019°, scanning rate 0.2 s/step, and scanning range 5°~70°.

The microstructure within each specimen was examined by the field-emission scanning electron microscope (SEM). The central section of each sample was extracted and analyzed after grinding and gold-spraying. The working voltage used in the field-emission SEM (Nova Nano SEM450, FEI company, Hillsboro, OR, USA) was 10.00 kV.

## 3. Results and Discussion

### 3.1. Effects of Four Curing Methods on the Mechanical Properties of Mortar

Figure 2 displays the compressive strength of each cement after aging for various durations under different curing methods. Figure 2a,b are the compressive strength data of hydration 1 day, 2 day and 3 day for each sample under different curing methods. It can be seen from the figure that metakaolin-incorporated specimens with 10% metakaolin (M10) have the highest compressive strength under the same curing period and curing methods. It can be seen from Figure 2a that under the curing method of 0# (60% humidity), the compressive strength of cement specimen M10 in 1 day can reach 15.61 MPa, which is 3.18 times that of ordinary Portland cement under the same curing method. The compressive strength is close to that of Portland cement for 3 days under the same curing condition. Similarly, the compressive strength of 3# M10 metakaolin-incorporated specimens after curing for 3 days is close to that of ordinary Portland cement under the same curing method (3# PC) for seven days (conformed by Figure 2c). It can be seen that metakaolin can significantly improve the early strength of ordinary Portland cement [24,25]. Figure 2a,b shows that the compressive strength of the 1#PC, 2#PC and 3#PC specimens in the first three days is significantly higher than that of the 0#PC specimen at the same age. As shown in Figure 2d, the 28 day compressive strength of PC specimen of 1#, 2# and 3#, was 42.37 MPa, 45.90 MPa and 43.08 MPa, respectively. Compared with that of the fully drying specimen (0#), the compressive strength of 1#PC, 2#PC and 3#PC specimens increased by 19.3%, 29.2% and 21.2%, respectively. This wholly confirms that ordinary Portland cement is highly sensitive to water, and the importance of early wet curing of Portland cement [8,10,13]. On the contrary, for the metakaolin-incorporated specimens under the four different curing methods, the early strength of the specimens has little difference, indicating that the addition of metakaolin can significantly reduce the sensitivity of silicate cement to water. Even in a dry environment, metakaolin cement can obtain high compressive strength. Figure 2 shows that the optimum dosage of metakaolin is 10%, and the M10 specimens in each group show the best mechanical properties. From the Figure 2d, we can see that the best curing method is 3# and the 28-day compressive strength of 3# M10 specimen can reach 49.12 MPa, which is about 1.16 times of that of ordinary Portland cement specimen under the best curing method (1# PC).

In construction engineering, the dual advantage of early demolding includes shortening the curing cycle of the workpiece and improving the productivity in practical application. From the result above, for ordinary Portland cement, the compressive strength should reach at least 14 day to stabilize, while for M10, the compressive strength after 7 day is sufficiently similar to the final strength. According to CB/T 70-2009 [26], the stent can only be removed when the sample is cured for 24 h at 20 °C; however, for sample M10, compressive strength requirement was achieved in less than one day, which would significantly shorten the time to demold each workpiece and greatly quicken construction projects that require cement.

From the particle size analysis of metakaolin (MK) and PC in Figure 1a, the particle size of MK was considerably smaller than that of ordinary Portland cement. Figure 1b,c shows the nitrogen adsorption and desorption isotherms, and the specific surface diagram of PC and MK. The specific surface area of MK is approximately 96 times larger than that of PC. Figure 3 depicts the filling effect of MK; MK increases the excess slurry above the cement mortar, which actually increases the compactness of the mortar [27]. Furthermore, the water film on the particle surface also thickens, which lubricates the particles, and thus, accelerates the reaction rate and extent, and improves the early strength. Due to the filling effect contributed by the small MK particle size, the cracks that typically form in cement specimens are largely filled, which improves the density and compressive strength of the workpiece. Moreover, the high pozzolanic activity enables the large number of Al_2_O_3_ and SiO_2_ within MK to react with Ca(OH)_2_ to generate a fibrous calcium silicate hydrate (C–S–H) gel, which further improves the compressive strength of the specimen [28,29]. However, if metakaolin exceeds 10% substitution, the compressive strength of the specimen instead declines substantially. This behavior can be attributed to the fact that if the substitution rate is excessive, the content of silicate within the system would be too low. This also translates to the content of both the generated Ca(OH)_2_ and the calcium silicate hydrate (C–S–H) gel, which is the primary contributor of the strength phase, would be insignificant. Lastly, the combination of a small particle size, large specific surface area, high water absorption, and high substitution rate of MK will increase the consistency, reduce free water content and extent of hydration, and ultimately weaken the mortar. The filling effect from the small particle size would not suffice to compensate for the strength loss caused by the incomplete hydration.

### 3.2. Effect of Four Curing Methods on the Durability of Mortar

#### 3.2.1. Compressive Strength after Sulfate Attack under Four Curing Methods

The four figures in Figure 4 above show the changes in compressive strength of each specimen after 28, 90, 120 and 180 days of erosion under four different curing methods. The compressive strength of the sample after sulfate attack will change with the extension of the erosion period. For those samples with incomplete hydration, loose internal structure and poor durability, the amplitude and rate of change of the compressive strength will fluctuate greatly. From Figure 4, M10 displayed the greatest compressive strength after 180 days of erosion under any of the four different curing methods. Moreover, the compressive strength of M10 exhibited the least fluctuation throughout the eroding period, which attests to its incredible stability and resistance to erosion and sulfate attacks. For all of the specimens, the compressive strength initially rises, then after a period of time, declines. This can be attributed to the large number of sulfate that initially reacts with the hydration products resulting in swelling and the formation of ettringite and gypsum, which moderately compacts the internal structure of the specimen and improves the strength. As erosion progresses, the internal stress caused by the volume expansion, increases. Simultaneously, sulfate calcium silicate gel begins to form, which whittles down the main strength phase and gradually form cracks in the specimen. In each curing methods’ group of specimens, the change range and rate of the strength of PC are the most obvious, indicating that PC is internally looser, possesses wider gaps and can be easily corroded or damaged [30,31,32,33,34]. Comparatively to PC, the strength of M10 increased and decreased to a lesser extent, which can be ascribed to the denser structure, more complete hydration, and higher sulfate resistance. The erosion is not serious and is not easy to be eroded. At the same time, no matter which kind of curing method is used, the M10 shows the highest compressive strength after 180 days of erosion, and the compressive strength of the sample 3#M10 can reach the maximum value of 41.62 MPa. Therefore, the optimal amount of MK is 10% (M10), when accounting for erosion due to sulfate. Despite dry conditions, the M10 specimens exhibit excellent strength and durability.

To see the changes of compressive strength of PC and M10 under various curing conditions more clearly and intuitively, we drew Figure 5. Figure 5 reveals that for the PC and M10, the best curing method is 1# and 3#. As can be seen from the Figure 5, for PC and M10 samples, compared with other curing methods, the variation range of compressive strength of sample 1#PC and 3#M10 in the whole erosion process fluctuates very little, and the maximum compressive strength reaches 37.98 MPa and 41.61 MPa respectively, after 180 days of erosion. In addition, the compressive strength of ordinary Portland cement samples changed rapidly after erosion, but that of M10 samples change little. Comparing 3#M10 to 0#M10, the strength increased by a mere 10.6% after 180 days of erosion, which is quite insignificant. In other words, for the M10 sample, the dry curing method (0#) and optimal curing method (3#) show little difference in sulfate erosion resistance. Therefore, adding MK can actually reduce the great dependence of PC towards water in the early stages. Therefore, in the dry areas of northwest China where wet or early wet curing may not be available, the loss of strength due to insufficient early wet curing conditions can be largely mitigated by adding MK to PC, optimally at 10%.

#### 3.2.2. Mass Loss Rate after Sulfate Attack under Four Curing Methods

The mass loss ratio of samples under differing curing methods is summarized in Figure 6. From Figure 6, the quality of all specimens under different curing methods increases first and then decreases. Briefly, sulfate permeates the specimen and reacts with hydration products to form ettringite and gypsum. Volume expansion from the formation of ettringite and gypsum fills the void and increases the compactness of the specimens. Moreover, the quality of all specimens are increasing, firstly. As the erosion progresses, cracks in the specimens begin to appear and then quality reduction gradually occurs until the quality loss is compared with the mass at 28 days. Similar to the compressive strengths, the rate and extent of the mass change of PC are the quickest and largest compared to the other specimens. However, the MK compound with a substitution rate of 10% (M10) shows the least mass change and the mass loss rate of 3#M10 is only 0.18% after 180 days of erosion, which attests to its superior stability and corrosion resistance.

To see the mass loss rate of PC and M10 under different curing conditions more clearly and intuitively, we drew the Figure 7. As shown in Figure 7, the longer the initial wet curing time, the better the sulfate resistance of the specimen is. The mass loss rate of 1#PC after 180 days of erosion was 0.57%, while the mass loss rate of 3#M10 was 0.18%, which is close to 0.14% of 1#M10. Therefore, the 3#M10 specimens showed a strong advantage in sulfate attack durability.

In order to more clearly see the relationship between the compressive strength and sulfate resistance of the sample, Figure 8 was drawn. As we have explained in the previous discussion, for metakaolin cement mortar, the 3# curing method is the best curing method, so Figure 8 shows the relationship between the compressive strength and the mass loss rate under the 3# curing method. Figure 8a shows the relationship between the initial compressive strength of the sample and the mass loss rate eroded for 180 days. Figure 8b shows the relationship between the compressive strength and the mass loss rate of the sample after 180 days of erosion. We can see that there are very similar changes in the two figures. Both the initial compressive strength and the residual compressive strength are negatively correlated with the mass loss rate. In other words, the greater the compressive strength of the specimens, the smaller the mass loss rate after 180 days of erosion, which indicates that the specimen’s resistance to sulfate attack is better. In the figure, we can also see that the overall performance of the sample M10 is the best.

### 3.3. X-ray Diffraction (XRD) Analysis

Figure 9a presents the XRD spectra of PC mortar specimens under the four different curing methods: 0#PC, 1#PC, 2#PC, and 3#PC. Comparing PC under dry curing conditions (0#PC) to PC under standard curing conditions (1#PC), 2#PC and 3#PC, the content of calcium hydroxide (C–H), hydrated calcium silicate gel (C–S–H), calcium aluminate hydrate (C_4_AH_13_) and ettringite (AFm) are significantly lower. Therefore, for PC specimens, the mechanical properties of 0#PC at each curing age are all smaller than those of other groups, which is consistent with the rule obtained above. This shows once again that adequate wet curing time in the early stage of hydration is very necessary. The lesser content can be ascribed to 0#PC lacking the recommended pre-wet curing, which resulted in an incomplete hydration [35,36]. Moreover, Figure 8b presents the XRD spectra of PC mortar specimens and M10 specimens under different curing methods: 0#PC, 0#M10, 3#M10, and 3#M10 erosion. Among them, the sulfate erosion age of the 3#M10 sample was 180 days. 3#M10 shows that there are fewer amounts of calcium hydroxide (C–H) and higher amounts of tetracalcium aluminate hydrate (C_4_AH_13_), ettringite (AFm) and hydrated calcium silicate gel (C–S–H) compared with other specimens 0#PC and 0#M10 displayed in Figure 9b. The result is due to the SiO_2_ and Al_2_O_3_ in MK that react with the hydration product and consumes the calcium hydroxide (C–H) in the system to generate AFm. However, higher amounts of calcium aluminate hydrate can be attributed to the fact that the hydration process of 3#M10 is more thorough and more calcium aluminate hydrate (C_4_AH_13_) and C–S–H are generated than that of 0#M10 and 0#PC. At the same time, ettringite crystals are the most abundant and calcium hydroxide has few in the sample system after 3#M10 sulfate erosion. This is because sulfate ions enter the sample and react with calcium hydroxide in the system, consuming calcium hydroxide and producing more ettringite.

### 3.4. SEM Analysis

As shown in Figure 10, the 0#M10 specimen is comprised of a small amount of scattered hexagonal, plate-like crystal calcium hydroxide (C–H), acicular prism crystal ettringite (AFm), and a large number of fibrous hydrated calcium silicate gels (C–S–H) [37]. Apparently, the surface is quite compact and without large voids. However, a small number of cracks can be seen. The surface morphology can be assigned to the incomplete hydration of the M10 cement under dry conditions. Figure 11 reveals a number of large macropores, and unreacted raw and clinker particles in 0#PC [38]. The specimen also contains a large amount of calcium hydroxide (C–H) and ettringite (AFm) generated by cement hydration. However, the surface does not seem compacted and is dispersed with a number of large cracks. As shown in Figure 12, the structural density of 3#M10 seems quite high and free of large pores. Apparently, there is an abundance of fibrous C–S–H gels and a scarcity of calcium hydroxide crystals on the surface of 3#M10, which can be ascribed to the reaction between calcium hydroxide, and SiO_2_ and Al_2_O_3_ in MK, to form the C–S–H gels [15,39,40]. As a result, the content of calcium hydroxide is very low, which attests to the completeness of the hydration, dense internal structure and very high strength and durability. In Figure 13, the eroded 3#M10 specimen manifests a loose internal structure due to erosion, as marked by the many pores and fine cracks. This can be explained by the large amount of sulfate that entered the specimen and consumed the calcium hydroxide, which reduces the alkaline environment in the specimen. As we know, proper alkaline environment has important resistance to sulfate attack in the Portland cement system. Concurrently, the erosion degenerates the calcium silicate gel which is the main strength phase in the specimen. Eventually, the volume expansion from the resulting ettringite and gypsum accumulates internal stress, which produces fine cracks and destroys the compactness and compressive strength of the specimen.

## 4. Conclusions

In this paper, the influence of the addition of metakaolin on mechanical properties and durability of Portland cement mortar was studied. The results show that: The filling effect of metakaolin (MK) can effectively increase the density of Portland cement (PC). Moreover, the large surface area and strong water absorption of MK will thicken the water film on the surface of the cement, which lubricates the particles, accelerating the reaction speed and extent, and the optimal dosage of metakaolin (MK) is 10%. The sample can significantly improve the compressive strength and durability of Portland cement mortar, especially the early strength. The compressive strength (at 28 day) and durability of MK10 are superior to that of the other samples.In this paper, four different curing methods were designed by changing the early wet curing time, and the influences of the four curing methods on each sample were explored. The results showed that: The traditional silicate-based PC is highly dependent on the wet curing time in the initial curing stages, and typically requires a wet curing time of at least seven days. However, the incorporation of MK can greatly reduce the dependency of PC to water. Moreover, the compressive strength of MK-incorporated cement (3#M10) that has been wet cured for as short as three days can reach 49.12 MPa after 28 days, which can greatly shorten the curing time that require humid conditions. Therefore, the addition of metakaolin can alleviate the problem of high maintenance cost caused by the long wet curing time of Portland cement products in the actual production. Especially in areas with low humidity and when the early wet curing time cannot be guaranteed, the addition of metakaolin has more practical production significance.The optimized amount of MK in cement was found to be at 10% (M10), which demonstrated an excellent, comprehensive performance. Under four different curing methods, the best condition for optimizing compressive strength involved the first three days: humidity 95%, followed by humidity 60% for remaining days (3#). From the microstructure of 3#M10, the surface was densely compacted and the hydration was more thorough. Regardless of the 28 day compressive strength, the sulfate resistance and mass retention rate were outstanding.

## Figures and Tables

**Figure 1 materials-15-02035-f001:**
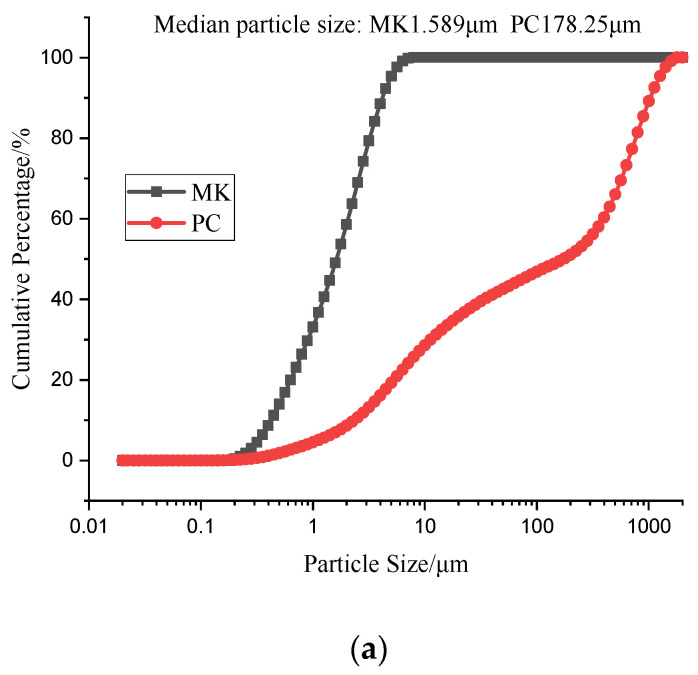
Physical properties of PC and MK. (**a**) Particle size distribution of PC and MK, (**b**,**c**) Nitrogen isotherms of PC and MK.

**Figure 2 materials-15-02035-f002:**
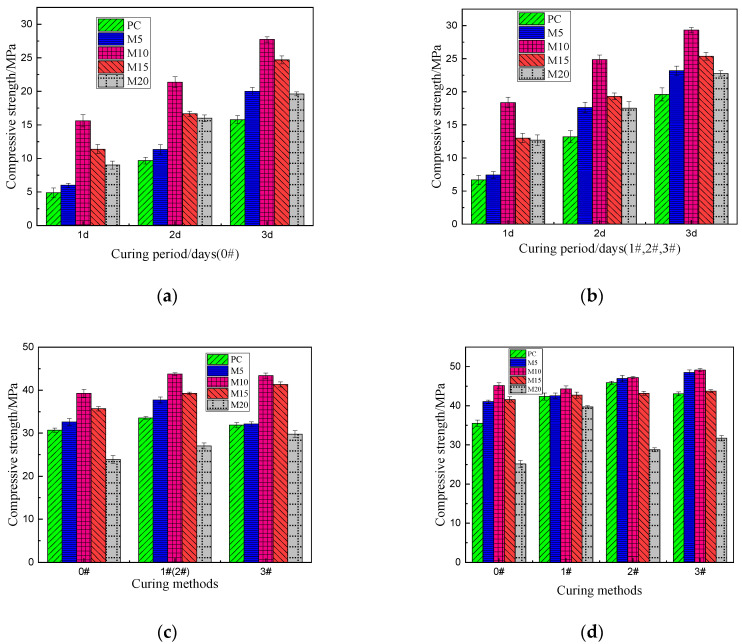
The compressive strength of cement after aging for 1, 2, 3, 7 and 28 days under different curing methods; (**a**) 1, 2, 3 days of 0#, (**b**) 1, 2, 3 days of 1#, 2# and 3#, (**c**) 7 days of 0#, 1#, 2# and 3#, (**d**) 28 days of 0#, 1#, 2# and 3#.

**Figure 3 materials-15-02035-f003:**
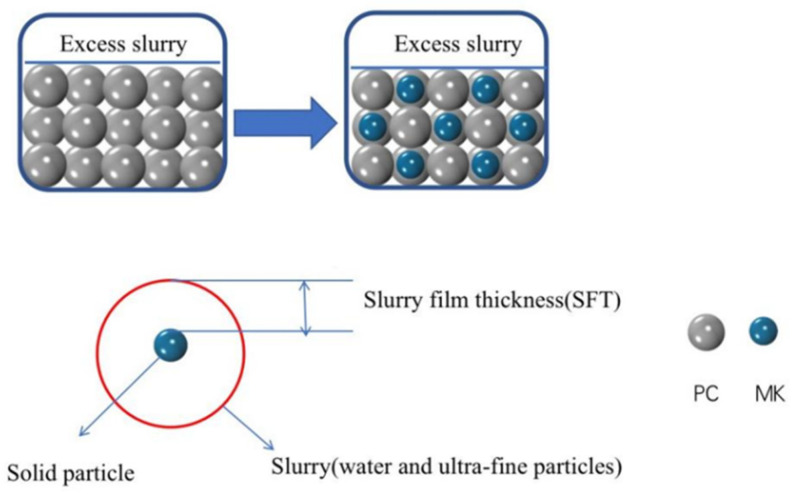
Principle Diagram of the Filling Effect of Metakaolin.

**Figure 4 materials-15-02035-f004:**
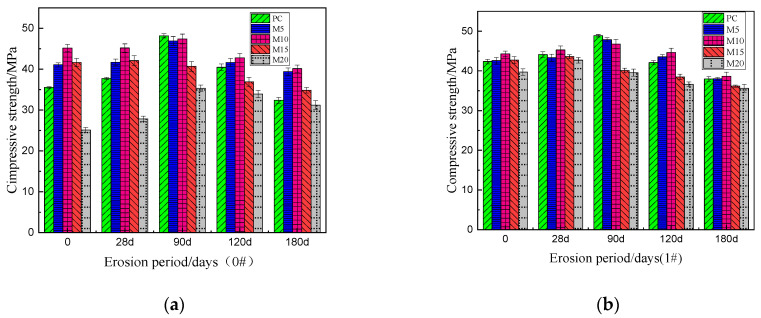
Compressive strength of cement mortar after a different erosion period under different curing methods; (**a**) 0#, (**b**) 1#, (**c**) 2#, (**d**) 3#.

**Figure 5 materials-15-02035-f005:**
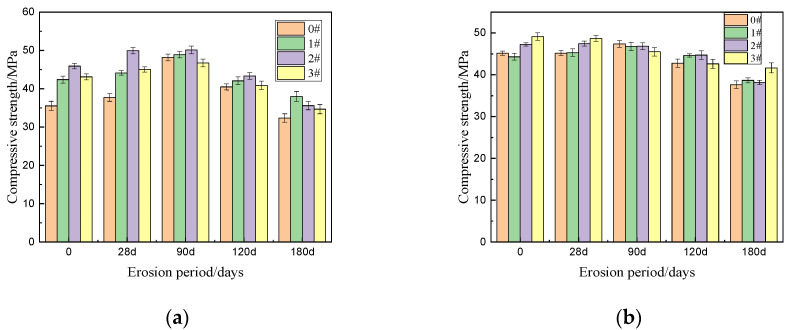
Compressive strength of PC and M10 after a different erosion period under various curing methods; (**a**) PC, (**b**) M10.

**Figure 6 materials-15-02035-f006:**
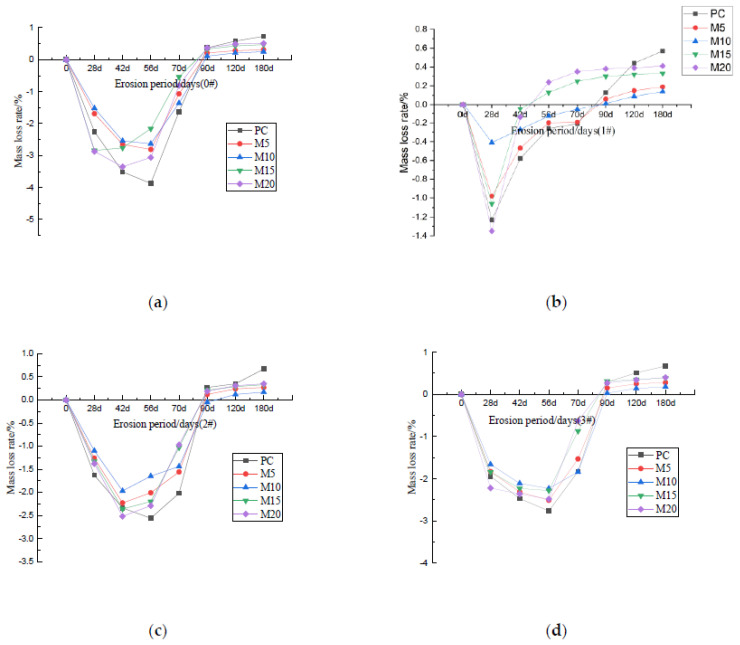
The mass loss rate of cement after a different erosion period under different curing methods; (**a**) 0#, (**b**) 1#, (**c**) 2#, (**d**) 3#.

**Figure 7 materials-15-02035-f007:**
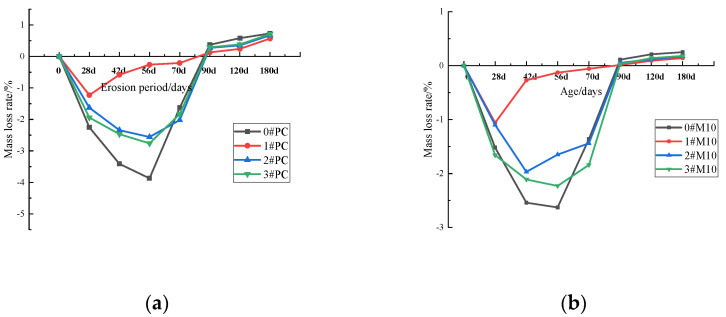
Mass loss rate of PC and M10 after a different erosion period under different curing conditions; (**a**) PC, (**b**) M10.

**Figure 8 materials-15-02035-f008:**
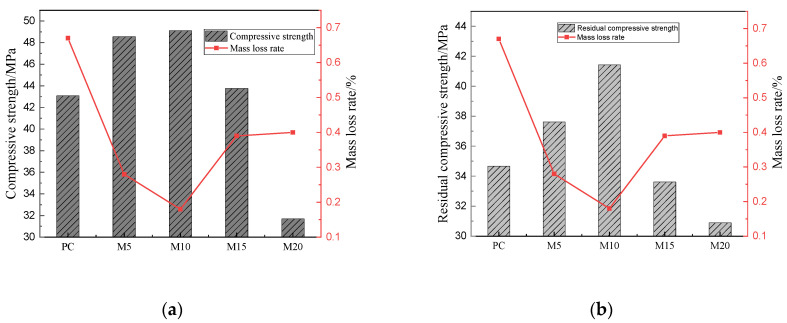
Relationship between compressive strength and mass loss rate; (**a**) Relationship between initial compressive strength and mass loss rate, (**b**) Relationship between residual strength and mass loss rate.

**Figure 9 materials-15-02035-f009:**
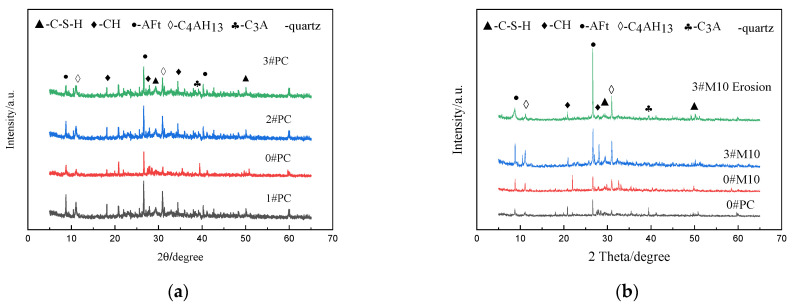
XRD patterns of PC and M10 after curing 28 days under various curing methods; (**a**) PC (0#, 1#, 2#, 3#), (**b**) PC (0#) and M10 (0#, 3#).

**Figure 10 materials-15-02035-f010:**
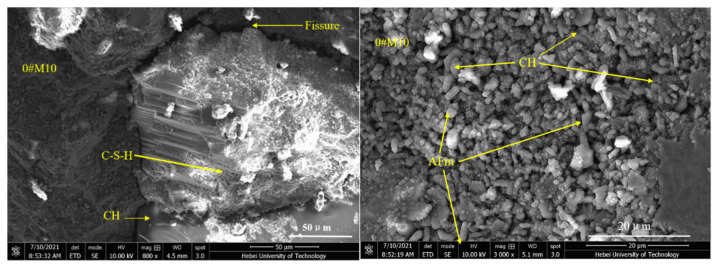
SEM images for 0#M10.

**Figure 11 materials-15-02035-f011:**
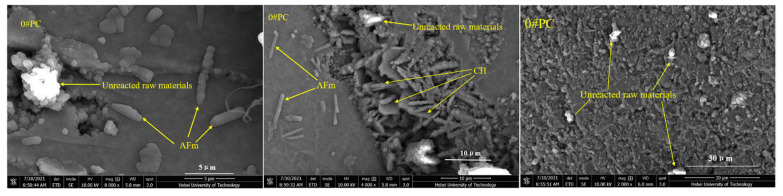
SEM images for 0#PC.

**Figure 12 materials-15-02035-f012:**
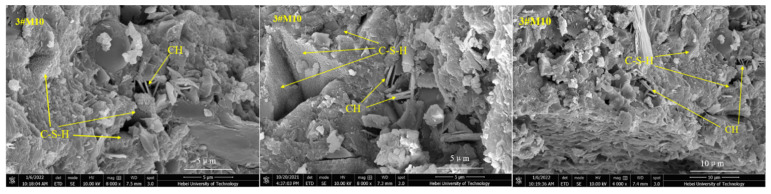
SEM images for 3#M10.

**Figure 13 materials-15-02035-f013:**
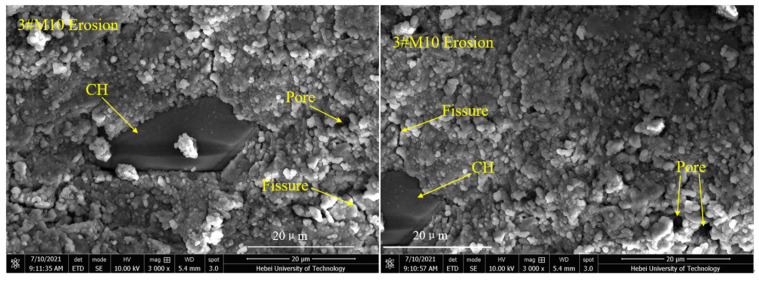
SEM images for 3#M10 erosion.

**Table 1 materials-15-02035-t001:** Chemical compositions of Portland cement and Metakaolin/%.

Material	CaO	SiO_2_	Fe_2_O_3_	Al_2_O_3_	MgO	SO_3_	TiO_2_	K_2_O	Others	LOI
Cement	49.06	26.56	2.58	10.37	6.33	1.80	1.53	0.828	0.942	1.62
MK	0.283	58.05	0.845	38.15	-	-	2.17	0.224	0.278	0.40

**Table 2 materials-15-02035-t002:** Physical properties of river sand.

Raw Material	Water Absorption/%	Specific Gravity	Fineness Modulus	Unit wt% (kg/m^3^)
River Sand	0.95	2.58	2.70	1588

**Table 3 materials-15-02035-t003:** Proportions of mortar mixture.

Mix	Samples	W/b	Cement (kg/m^3^)	MK (kg/m^3^)	Sand (kg/m^3^)	Water (kg/m^3^)	Superplasticizer (kg/m^3^)
1	PC	0.45	595.18	0	892.76	267.84	5.95
2	M5	0.45	565.42	29.76	889.83	267.84	7.14
3	M10	0.45	535.66	59.52	886.90	267.84	8.33
4	M15	0.45	505.9	89.28	883.96	267.84	9.52
5	M20	0.45	476.14	119.04	881.03	267.84	10.71

## Data Availability

The study did not report any data.

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
