# Peer review of "Investigating the Mechanical Properties and Durability of Metakaolin-Incorporated Mortar by Different Curing Methods"

_materials, 2022, doi:10.3390/ma15062035_

Round 1

Reviewer 1 Report

The authors have clearly indicated the novelty of their research and have improved different aspects of their paper in comparison with their first submission. The following are required to be considered by the authors before the paper being accepted. 

  • There are some typo mistakes in the paper that need correction.
  • The current conclusions needs to be amended as it is not presented properly. Therefore, authors are firstly required to state the main aim of the current research at the conclusion and then listing the main findings.

Reviewer 2 Report

Some comments have been addressed properly while others remain unanswered.

  • L32: The statement is still not clear. Consider rephrasing rather than adding a supplementary statement following it.
  • L99: Why was the strength of the mortar of 42 MPa selected? Is this a standard strength value for a specific application?
  • Table 3: The mix design values are incorrect. The total volume does not add to 1 m3. Consider using the ACI 211.1 absolute volume theory to check the values.
  • The comment: "Correlations between results of sulfate attack and compressive strength are needed with more details" has not been addressed. What is needed is how does the initial compressive strength affect the sulfate resistance of the mixes? Can a function be developed to correlate the strength to the mass loss due to sulfate attack?

Reviewer 3 Report

The content of the paper is understandable, except for my question in remark 2.

Remarks:

1. It would be very useful List of Abbreviations as MK, SEM, PC. They seem to be obvious for the Authors, but not every reader can know them immediately.

2. Could you explain, why different ratio of the humidity was applied 
for different samples? It is any rule? For sample #2 it was changed after 7d, but for sample #3 after 3d? Maybe I misunderstood something.
Moreover, it is not clear, whether every sample was subjected
to four different curing methods or sample 1 was under 0#,
sample 2 - under 1#. Total number of samples equals 5, number of curing method is 4.
If needed, please clarify this query even in the text of the manuscript
(see Tab 3 and lines 123-129).

Round 2

Reviewer 2 Report

All comments have been addressed.

This manuscript is a resubmission of an earlier submission. The following is a list of the peer review reports and author responses from that submission.

Round 1

Reviewer 1 Report

The paper examines the impact of MK and different curing methods on the compressive strength and sulfate attack resistance of cement mortars. The paper requires major revisions before it can be of publication quality. The following comments should be addressed: 

  • L29-30: "Due to the early strength and late strength of ordinary Portland cement is very low, so many scholars are trying to improve the mechanical properties of Portland cement." What do the authors intend to say from this sentence? A comment is necessary.
  • The introduction lacks a comprehensive review of literature relevant to concrete curing. Several recent papers have studied the effect of curing on different concrete materials. The authors should refer to these papers in the introduction and throughout the article for comparative analysis, where needed: https://doi.org/10.1016/j.conbuildmat.2016.02.007; https://doi.org/10.1080/21650373.2021.1883145; https://doi.org/10.1016/j.conbuildmat.2021.123963; https://doi.org/10.1016/0008-8846(94)90124-4
  • The last paragraph in the introduction should state the novelty and significance of this work while highlighting the experimental program.
  • L62: Properties of the river sand, including specific gravity, absorption, density, size distribution, among others should be mentioned.
  • L73: On what basis was this mortar mix design chosen? Is there any past literature or trial mix to validate this choice?
  • L78: What was the design workability to validate the increase in superplasticizer content?
  • Table 2: It is common practice to report the mix design in kg/m3. Consider revising.
  • L90: On what basis were these curing methods chosen? How was 60% humidity achieved? Why this specific value was chosen?
  •  L99: How many samples were tested per mix and per test? 
  • L103: It is not clear whether samples were tested for durability after the designated curing methods or not? The text shows that they were all cured at constant temperature and humidity for 28 days. Then, these results are not related to the compressive strength results.
  • L120: Consider revising the number of significant digits for compressive strength values.
  • L139: Interesting approach. However, the study here is about mortars while this statement refers to concrete. Please elaborate.
  • L166: Sulfate attack results do not contribute to the knowledge in the field. Many past researchers have investigated the effect of MK on sulfate attack resistance of cement mortars. It is only interesting to see what the new curing methods could offer in terms of improvement to sulfate attack resistance.
  • L198: Mass loss rate related to sulfate attack. It should not be a new section but a sub-section.
  • Correlations between results of sulfate attack and compressive strength are needed with more details.
  • Figure 8 is too small. The compounds can be barely seen.
  • It would be interesting to show the effect of sulfate attack on the XRD spectra of at least one sample.
  • The annotation in the SEM images is not clear. Consider using different colors for the text.
  • How could the authors identify these compounds in the SEM? EDX/EDS spectra of a few compounds should be included.
  • It is recommended to include the amount of water that could be saved by shortening the curing of cement mortars made with MK.

Reviewer 2 Report

Dr. Yang's paper is well written, and the Authors were very clear about the practical problem faced. However, some aspects of the document require improvement. Between them:
- Check text around Table 2 (Lines 84 to 89)
- Is the target of XRD equipment Copper? Add the info at line 110.
- Add error bars to Figures 2, 4, and 5. Then check if these figures' values are statistically different and introduce this discussion into your text.

Reviewer 3 Report

Although the paper is well structured, and it is within the scope of the journal. However, the topic has been investigated extensively by different researchers around the world and there is no novelty of the current paper therefore, I must reject it.